# Interplay of valley polarized dark trion and dark exciton-polaron in monolayer WSe₂

Xin Cong[1], Parisa Ali Mohammadi [1], Mingyang Zheng[1], Kenji Watanabe [2], Takashi Taniguchi [3], Daniel Rhodes [4] & Xiao-Xiao Zhang [1] ✉

The interactions between charges and excitons involve complex many-body interactions at high densities. The exciton-polaron model has been adopted to understand the Fermi sea screening of charged excitons in monolayer transition metal dichalcogenides. The results provide good agreement with absorption measurements, which are dominated by dilute bright exciton responses. Here we investigate the Fermi sea dressing of spin-forbidden dark excitons in monolayer WSe₂. With a Zeeman field, the valley-polarized dark excitons show distinct p-doping dependence in photoluminescence when the carriers reach a critical density. This density can be interpreted as the onset of strongly modified Fermi sea interactions and shifts with increasing exciton density. Through valley-selective excitation and dynamics measurements, we also infer an intervalley coupling between the dark trions and exciton-polarons mediated by the many-body interactions. Our results reveal the evolution of Fermi sea screening with increasing exciton density and the impacts of polaron-polaron interactions, which lay the foundation for understanding electronic correlations and many-body interactions in 2D systems.

Excitonic physics in 2D semiconductors of monolayer transition metal dichalcogenides (TMD) has attracted great attention for over a decade[1]. This 2D system hosts exceptionally strong excitonic interactions, different excitonic complexes, and rich valley and spin degrees of freedom[1,2]. In the presence of free charges, a new excitonic resonance appears below the neutral exciton energy. This charged excitonic resonance has been initially assigned to the three-body trion state, in which an additional charge binds with an exciton[3]. Recent experimental and theoretical studies reveal that the interactions between the charge-neutral excitons and free carriers mimic the Fermi-polaron responses[4–13], especially at a higher doping density, where an exciton interacts not only with a single-particle free charge but with Fermi sea fluctuations. At a low doping density, many-body interactions between charges and excitons are not strong, and the initial neutral exciton and trion modeling are sufficient. As doping increases, neutral excitons and trions are expected to go through a smooth transition into repulsive and attractive exciton-polarons, accompanied by blueshifts and redshifts in energy. These energy shifts, as well as the evolution of the excitonic oscillator strength measured from the bright excitons up to moderate doping, agree with the theoretical calculations using the exciton-polaron picture[5,8,13].

The Fermi-polaron model treats the quasiparticle responses of a single mobile impurity in a surrounding Fermi sea[14–16]. In comparison, the exciton density in monolayer TMD can be tuned by laser fluence and be comparable to or exceed the charge density, where the analogy to a single mobile impurity no longer applies. The modification to Fermi sea screening at high exciton densities, however, is still not well understood. Apart from the bright excitons previously studied in reflection contrast measurements[4,6,7,11,12], different spin and momentum dark exciton species have been established, which are also expected to have many-body interactions with charges. The coupling between these different species of exciton-polarons has not yet been experimentally investigated.

[1]Department of Physics, University of Florida, Gainesville, FL, USA. [2]Research Center for Functional Materials, National Institute for Materials Science, 1-1 Namiki, Tsukuba, Japan. [3]International Center for Materials Nanoarchitectonics, National Institute for Materials Science, 1-1 Namiki, Tsukuba, Japan. [4]Department of Materials Science and Engineering, University of Wisconsin-Madison, Madison, WI, USA. ✉e-mail: xxzhang@ufl.edu

Here we study the trion to exciton-polaron crossover for spin-forbidden dark excitons in a WSe$_2$ monolayer and further reveal the interactions between valley-polarized dark polarons. The spin-forbidden dark excitons correspond to the spin ±1 energy ground-state excitons in tungsten-based monolayer TMD due to the opposite spin alignment between the bottom conduction band and the top valence band[17–19]. Optical detection of these dark states is possible by applying an in-plane magnetic field[19–21] or through a weak out-of-plane dipole emission in photoluminescence (PL)[18,22,23]. Compared to the optically-allowed bright excitons, these dark excitons hold much longer exciton and valley lifetimes due to the lack of a rapid radiative channel and the absence of intervalley exchange interactions[19,24]. In this report, we extracted and analyzed the Zeeman-split dark exciton PL at the p-doping side with Fourier plane imaging spectroscopy. A distinctively different doping dependence was observed for the K and K' valley dark trions as the doping increased across a critical density, which corresponds to the onset of strong Fermi sea screening effects. We characterized the shifting of the critical density as the exciton density increases with pulsed excitation and extracted the modification to polaron screening from the dilute to high exciton density. The exciton dynamics for the different valley-polarized dark trions were further examined with time-resolved PL. Combined with valley-selective excitation measurements, we reveal the many-body interactions between the K and K' valley dark trion and exciton-polaron mediated by the Fermi sea fluctuations.

## Results

### Two-segment gate dependence of dark trion PL

A low excitation density is first used to investigate the dark exciton PL in the dilute exciton limit. Figure 1a shows the gate-dependent PL from an hBN-encapsulated monolayer WSe$_2$ device at 4 K with a pulsed laser

of 1.88 eV and 0.6 μJ/cm$^2$ fluence (see "Methods"). The excitation laser was linearly polarized unless otherwise specified. The different peak features have been characterized in previous studies and are associated with bright exciton, dark exciton, phonon-assisted emissions, and multi-exciton complexes[9,25,26]. The spin-forbidden dark excitons have a weak out-of-plane dipole and in-plane emission, while bright excitons and dark phonon replicas have an in-plane dipole and out-of-plane emission. To explicitly examine the optically dark excitons, we used the Fourier plane imaging spectroscopy to obtain PL spectra with both photon energy and emission momentum resolutions[27]. The dark exciton contributions can be extracted by comparing the PL's different momentum distributions (see Supplementary Information for the subtraction procedure). Figure 1b shows the extracted dark excitons within a zoomed-in gate voltage range. The peak at ~1.69 eV is the neutral dark exciton D, with the n-type and p-type doped dark trions D$^-$ and D$^+$ observed at the positive and negative gate voltages, respectively.

Under an out-of-plane magnetic field, the Zeeman-split p-type dark trions show distinctly different doping dependencies, as shown in Fig. 1d. The corresponding g-factors of the neutral dark exciton and p-type dark trions from their energy splitting are ~10, consistent with previous reports[28]. The shifted electronic bands at the K and K' valleys under the magnetic field[28] are sketched in Fig. 1c. Within the charge neutral regime, the higher energy emission D1 excitons reside in the K' valley and the lower energy D2 excitons are in the K valley following the Zeeman energy shifts, as labeled in Fig. 1c, d. The p-type dark trions have the intervalley configuration as the lowest energy configuration, and excitonic structures of the corresponding D1$^+$ (D1 dark exciton in K' valley + K valley hole) and D2$^+$ (D2 dark exciton in K valley + K' valley hole) are plotted in Fig. 1c. The PL signals of the D1$^+$ (D2$^+$) states come from the recombination of the D1 (D2) exciton while leaving behind the

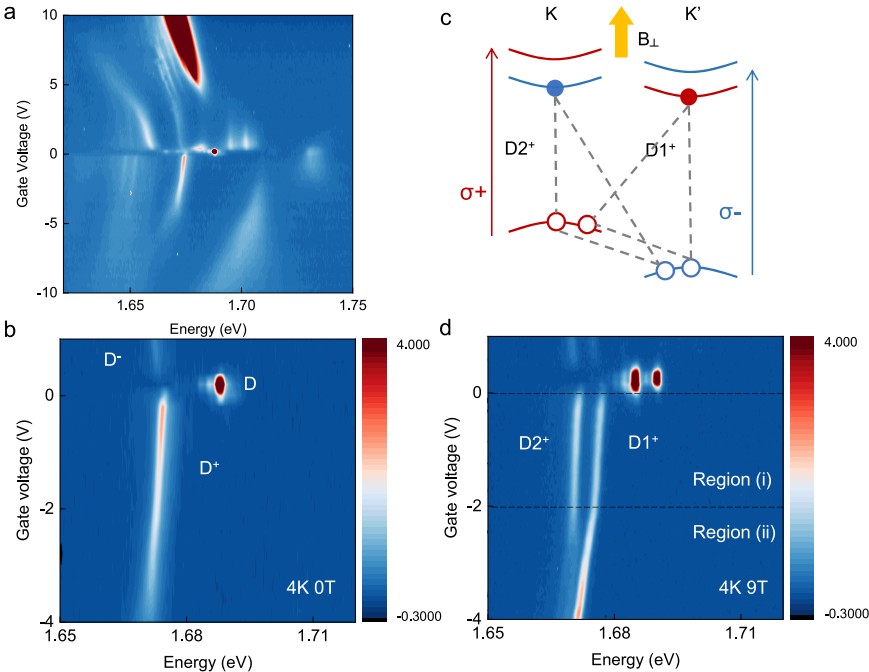

**Fig. 1 | Gate-dependent dark exciton PL. a** Gate-dependent photoluminescence (PL) of an hBN-encapsulated monolayer WSe$_2$. The charge neutral point is near 0 V. **b** The spin-forbidden dark exciton PL spectra extracted from Fourier-plane imaging are shown in a zoomed-in gate voltage range. D corresponds to neutral dark exciton. D$^+$ and D$^-$ correspond to p-type and n-type dark trions. **c** The schematics of electronic bands near K and K' valleys under an out-of-plane magnetic field in a monolayer WSe$_2$. σ+ (σ−) polarized light creates direct optical excitation (bright exciton) in the K (K') valley. The experimental assignment of the valley index and circular polarizations are shown in Supplementary Information Section 3. In the p-doped regime, dark excitons bind with a hole in the other valley to form an intervalley dark trion. D1$^+$ and D2$^+$ indicated the dark trions in the K' and K valley, respectively. **d** The dark exciton PL spectra under a 9 T out-of-plane magnetic field, with otherwise the same conditions as in (c). D1$^+$ and D2$^+$ and the Zeeman-split dark trions as denoted in (c). The dashed line at ~ −2V marks the crossover between region (i) and region (ii).

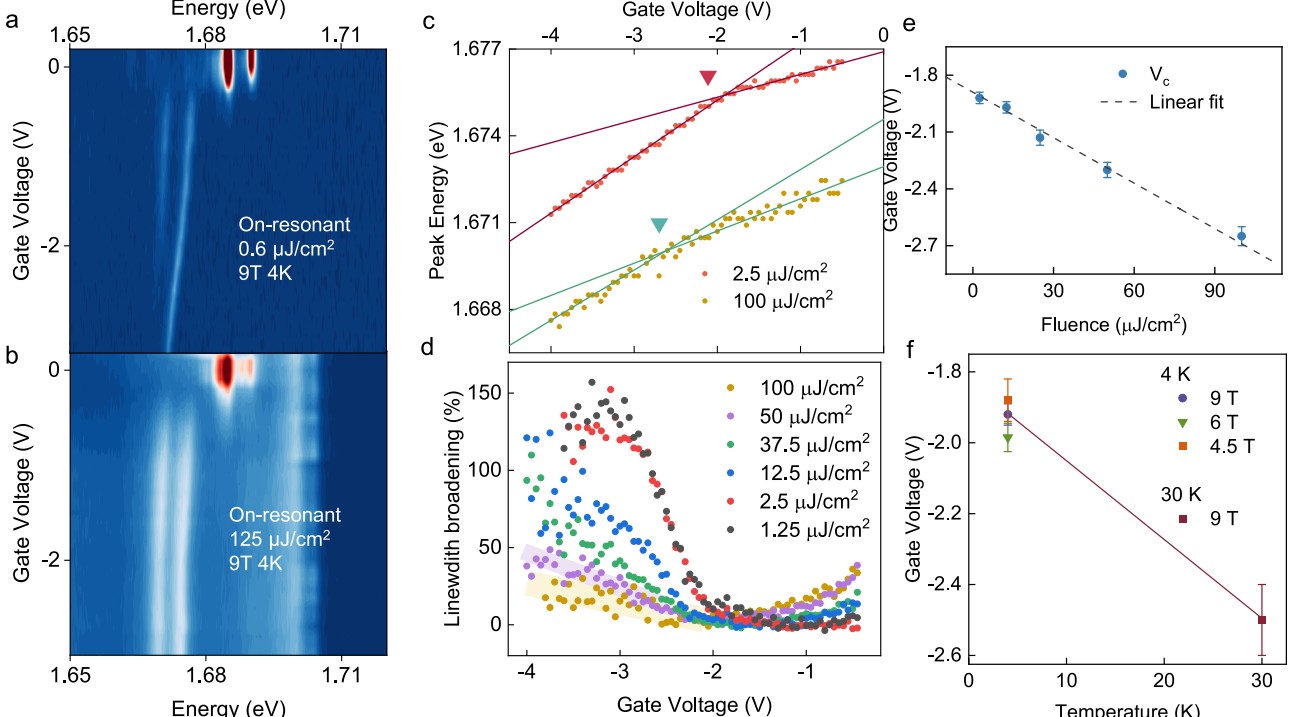

**Fig. 2 | Fluence-dependent trion to exciton-polaron crossover. a** Dark exciton gate dependence with low fluence excitation condition taken at 4 K, 9 T. The pulsed excitation is chosen to be resonant with the bright neutral exciton at -1.72 eV. PL spectra were taken with a long pass filter at -1.69 eV to filter out the excitation laser. **b** Same measurements as in (**a**) taken with a high excitation fluence. Throughout the same gate voltage range, D1⁺ and D2⁺ remain in region (i). **c** The D1⁺ peak energy (with pulsed 1.88 eV excitation) at low and high fluences are plotted to show the extraction and the shift of $V_c$. The 2.5 μJ/cm² data are displayed vertically for clarity. The D1⁺ energy shift can be fitted with a two-segment linear dependence on the gate voltage. The crossing of the two linear functions, denoted by the inverted triangle, is taken as the $V_c$. **d** The percentage linewidth broadenings of the D2⁺ state are compared for different fluences, with pulsed 1.88 eV excitation. The fitted results for 50 and 100 μJ/cm² have higher uncertainty because of the overall broadening of the PL signals at high fluences. The shaded areas are guides to eyes. **c**, **d** are both taken at 4 K, 9 T. **e** The fluence dependence of $V_c$ as extracted from D1⁺ peak energy shifts, which can be fitted with a linear function, indicated by the dashed line. **f** With the low fluence condition (2.5 μJ/cm², pulsed 1.88 eV), the extracted $V_c$ at different out-of-plane magnetic fields at 4 K, and at 30 K, 9 T.

hole in the K (K′) valley. When increasing hole doping, additional hole carriers will first populate the K valley valence band and, therefore, should favor the formation of D1 trions, D1⁺, if the two trions are in thermal equilibrium. In contrast, D1⁺ and D2⁺ show clear two-segment gate dependencies as the gate voltage $V_g$ reaches a crossover voltage $V_c$ (~ −2 V), labeled as region (i) and (ii) in Fig. 1d. In region (i), D1⁺ and D2⁺ have similar amplitudes, linewidths, and their peak energy redshifts are both around 0.8 meV/V. As the doping increases into the region (ii), D2⁺ shows a decrease in amplitudes as its linewidth broadens. On the other hand, the emission intensity increases for D1⁺, and its peak energy has a larger redshift as the doping density increases, with a slope of -2.0 meV/V. In addition, D1⁺ shows a slight linewidth narrowing within this range. With further hole doping beyond region (ii), D1⁺ intensity also decreases while the bright trion PL intensity increases. The fitted amplitudes and linewidths of D1⁺ and D2⁺, as well as the data with higher dopings are included in the Supplementary Information (SI) Sections 2 and 4, and Figs. S2, 3 and 6.

The two-segment doping dependence of dark trions in Fig. 1d can be first qualitatively understood by considering the trion and exciton-polaron responses. At very low doping density (near zero $V_g$), the charge density in the K and K′ valleys are both small. As dark excitons have a long exciton lifetime, dark excitons in both valleys can scatter and bind with an additional charge in the other valley to form the D1⁺ and D2⁺ trions despite the low doping density. The thermal non-equilibrium between two dark trions is expected due to the lack of intervalley exchange, which can lead to their similar amplitudes despite the valley charge imbalance. At a higher doping density, the K valley valence band will be filled to high enough doping density such

that the modified Fermi sea screening (Fermi-polaron or Suris tetron[29]) gives rise to a larger exciton energy redshift, corresponding to the increased redshift of D1⁺ in region (ii). The slope increase in the excitonic energy redshift from the region (i) to (ii) also qualitatively agrees with the expected slope changes during the trion to exciton-polaron crossover[8,13,30].

## Fluence and magnetic field dependence of the crossover behavior

The crossover point, $V_c$, that defines region (i) and region (ii) can be shifted by the exciton density, which also confirms our assignment of trion to exciton-polaron crossover. Figure 2a, b demonstrates the contrasting behaviors with different exciton densities using an on-resonance pulsed excitation. The doping level at −2 V is estimated to be around $1.4 \times 10^{12}/\text{cm}^2$ with the device geometry and the hBN gating dielectric thickness using the geometric capacitance. The low exciton density ($<1 \times 10^{11}/\text{cm}^2$) scenario in Fig. 2a has consistent doping dependence as shown in Fig. 1d. Measurements done with alternative continuous wave laser (1.96 eV, 05−50 μW) yield the same results as Fig. 2a, as the exciton densities were low to remain in the dilute exciton limit. With a high exciton density of $-1 \times 10^{13}/\text{cm}^2$ (estimated from bright exciton absorption) shown in Fig. 2b, the gate-dependent PL signals remain in the region (i) throughout the same gate voltage range. We extract this crossover behavior as a function of exciton density by tracing the D1⁺ peak energy and slope changes as a gate voltage function. A pulsed 1.88 eV laser is used for fluence-dependent measurements below to avoid significant changes in absorption amplitudes as a function of gate and fluence. As shown in Fig. 2c, the

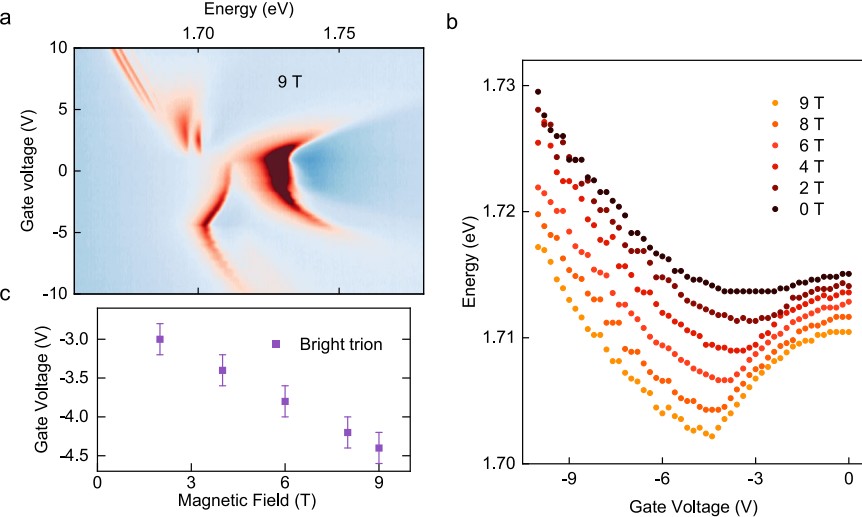

**Fig. 3 | Gate-dependent reflection contrast spectroscopy. a** Gate-dependent white light reflection contrast spectroscopy of the monolayer WSe$_2$ device at 4 K, 9 T. Landau level can be observed at high n-doped and p-doped regimes. **b** The peak energy of the p-type bright trion as a function of gate voltages at different out-of-plane magnetic fields. **c** The crossover voltages when the p-trion energy changes from redshift to blueshift with increasing doping density are plotted as function of the magnetic field. The 0 T field data are omitted due to the relatively large error bar.

D1$^+$ peak energy can be fitted with two-segment linear functions for the low and high doping regimes with a crossover point at $V_c$. The data for 2.5 and 100 μJ/cm$^2$ are plotted and vertically displayed for clarity, showing a shift of $V_c$ from −1.92 to −2.65 V. The fitted region (ii) slope for the high fluence data is smaller than that of lower fluence, which is also consistent with expected charge screening effects at an elevated exciton density. Figure 2e plots the extracted $V_c$ as a function of the measured fluences. As the exciton density increases with increasing laser fluence, there is an increase in $V_c$ amplitude, which can be fitted to a linear function of the gate voltage with a slope of −8 meV per (μJ/cm$^2$). The dark exciton density can be evaluated using 1% absorption of the fluence, which gives $3.3 \times 10^{12}$/cm$^2$ for 100 μJ/cm$^2$ fluence, and 1 V of gate voltage change is estimated to provide ~$0.7 \times 10^{12}$/cm$^2$ change in doping density. The fitted slope thus indicates the crossover requires ~0.2 hole per extra exciton.

The crossover onset can be alternatively extracted from the broadening of the D2$^+$ peak, which shows a consistent gate dependence as the D1$^+$ energy slope changes. The percentage broadening of the D2$^+$ compared to the narrowest linewidth for different fluency is plotted in Fig. 2d. The onset voltages extracted from the linewidth (see SI) provide a similar slope of the $V_c$ as a function of exciton density as the previous method (Fig. 2e). Here, we emphasize that the onset of D1$^+$ slope change and D2$^+$ broadening is similar only for linearly polarized excitation and, therefore, with equal K and K' valley excitations. We compare the distinctions with valley excitations and further discuss the D2$^+$ dynamics in the latter part of the report and in SI Section 3. Notably, the crossover $V_c$ is not sensitive to the magnetic field strength, as shown in Fig. 2f, if the Zeeman splitting is sufficient to lift the energy degeneracy of the two dark trions with minimal spectral overlaps. This reveals that the observed crossover does not originate from the shifting of the Fermi level when it touches both Zeeman-split valence bands, in contrast with the magnetic field dependence in the bright exciton energy kinks in absorption[31] shown in Fig. 3. At the observed $V_c$ ~−2 V in the dilute exciton limit, the Fermi level is also well below both valance bands and corresponds to a doping density of $1 \times 10^{12}$/cm$^2$ in the K valley and $4 \times 10^{11}$/cm$^2$ in the K' valley (taking $g_v = 12$[28] as the Zeeman splitting between the two valence bands). On the other hand, the crossover point $V_c$ shows a significant increase when the temperature is higher (see Fig. 2f). This can be qualitatively understood by considering the thermal excitations of carriers at higher

temperatures, which then require a higher doping to reach the same critical doping density. The $\Gamma_5$ phonon replicas of D1$^+$ and D2$^+$ also exhibit consistent gate dependence (see SI Section 4 and Fig. S6), in which we can observe the D2$^+$ phonon replicas eventually show an increase in energy redshift at even higher gate voltage when there are sufficient carriers in both valleys.

Combing the above fluence dependence measurements, our results infer that, at the dilute exciton limit, a crossover to a strongly modified Fermi sea screening occurs at a critical density of $n_c = (1 \pm 0.1) \times 10^{12}$/cm$^2$. This critical density corresponds to the carrier density in the K valley, and give rise stronger screening of D1$^+$, following the band schematics in Fig. 1c. Once reaching the $n_c$, an further increase of excitons density will require adding more charges to recover a similar screening effects. The extracted shift of 0.2 hole/exciton from Fig. 2e serves as a lower bound estimation. Considering the longer dark exciton propagation distance and the potential over-estimation of dark exciton density, the estimated shift can be 1–2 hole/exciton. The crossover at such a critical density is not expected within the current exciton-polaron picture, which predicts a smooth transition between the trion and polaron at a much lower density[5,8,13]. Other crossover mechanisms including the transition between trion-hole complexes and exciton-polarons, and the development of rotons have been discussed in quantum wells[32] and TMDs[33,34], which predicts a comparably high crossover doping density. However, those discussions were focused on bright excitons and absorption signals. The dark excitonic states do not show observable absorption signals and we cannot make a direct comparison. A microscopic understanding of $n_c$ is beyond the scope of this report and will require further theoretical considerations.

Dark excitons are generated through bright exciton absorption and subsequent phonon-assisted relaxations, but this crossover behavior does not originate from the bright excitons' gate dependence. Figure 3a shows the corresponding white-light reflection contrast measurement of the sample. With a 9 T out-of-plane field, we can observe Landau levels at high n- and p-type dopings, consistent with previous reports[10,31,35]. At the p-doped side, the kink in the bright trion energy has been assigned to the relative position of the Fermi level compared to the two valence bands[31]. The doping dependence of the bright trion absorption signal therefore has a strong magnetic field dependence, which is distinctly different from that of the dark exciton

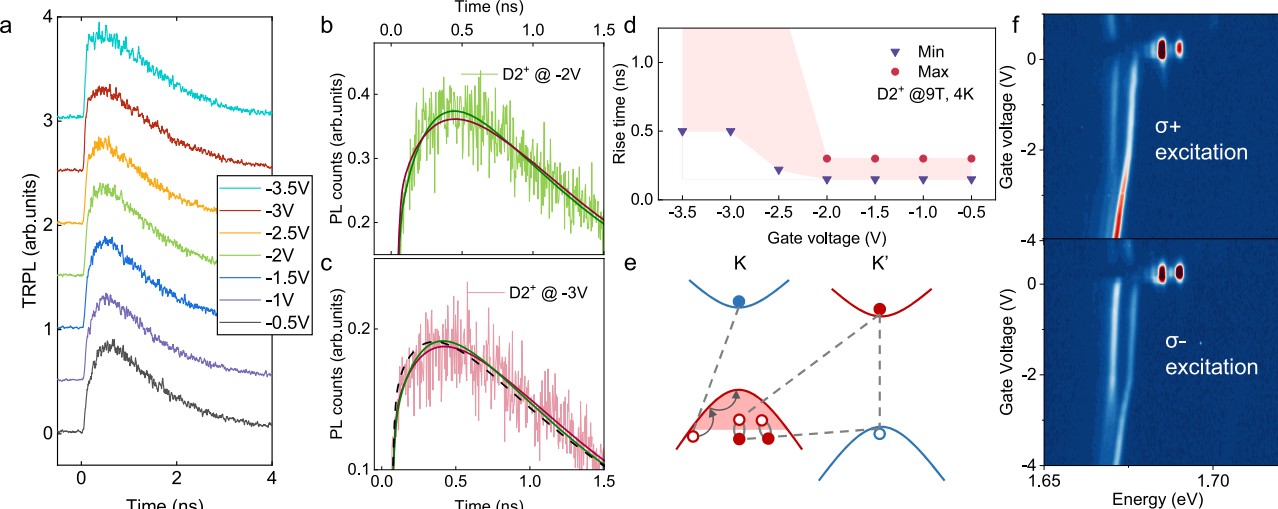

**Fig. 4 | D2⁺ dynamics and intervalley interactions. a** Time-resolved PL of the D2⁺ peak different gate voltages. **b**, **c** are the zoomed-in view of the D2⁺ rising edge at −2V and −3V. The solid red and green lines are the fitted dynamics with a 1 ns and 0.2 ns rise time, respectively, and they are plotted to contrast the different rise times. The black dashed line in (**c**) is the dynamics with zero rise time, which does not agree with the data. **d** The fitted rise time of D2⁺ as a function of the gate voltage. The shaded area denotes the error bar of the time scales. At $V_g < -3V$, the rise time is comparable to the decay time, making the upper bound estimation inaccurate. **e** Above $n_c$, Fermi sea in K valley interacts with the K' valley dark exciton through Fermi sea fluctuations. The same Fermi sea contributes to the K valley dark trion recombination, which correspondingly shows a possible blockage in emission/formation. The K' valley hole that binds with the K valley dark exciton is omitted for schematics clarity. **f** Gate-dependent PL spectra of dark trions (at the low fluence limit, 4 K and 9 T) are shown with different circularly-polarized and valley excitations. σ+ excitation favors the formation of D1⁺ trion and σ− favors the formation of D2⁺.

crossover behavior (Fig. 2f). We extracted the magnetic field dependence of the bright trion absorption "kink" by taking the gate voltage for the p-type trion energy shifts from a redshift to blueshift (see Fig. 3c). Indeed, the "kink" transition voltage plotted in Fig. 3b shows a linear dependence of 0.2 V/T. Taking the estimated doping density and the hole effective mass as -0.4 $m_e$[36], the extracted Fermi level shift yields a g-factor of -12 for the K and K' valence bands, agreeing with the reported value[28]. We also note that the onset of the "kink" does not occur as the Fermi level is shifted down and touches the K' valley (following the schematics in Fig. 1c), but instead occurs after the doping of the K' valley reaches $-1.2 \times 10^{12}/cm^2$, a doping density consistent with the above extracted critical density from the dark excitons.

**Many-body interactions between valley polarized dark trions**
Finally, we focus on analyzing the gate dependence of the D2⁺, which shows a quenching at higher hole dopings, opposite to the conventionally expected behavior of charged excitons. We measured the D1⁺ and D2⁺ exciton lifetimes with time-resolved PL (TR-PL), in which we selectively probed the emission dynamics with large momentum near the edge of the Fourier imaging plane to exclude the contribution from different bright states near the dark exciton energies. These dark trion lifetimes within this gate voltage range were measured to have a decay time of -1 ns and therefore reach quasi-thermal equilibrium within each excitonic species. Figure 4a shows the TR-PL measurements of the D2⁺ peak (12.5 uJ/cm² excitation) as a function of the gate voltages. The exciton rise time of D2⁺ showed significant changes across $V_c$ (~ −2 V), while negligible changes were observed in the corresponding dynamics of D1⁺ (see SI). The rise dynamics can be fitted with two exponential rise functions, with a fast (<10 ps) and a slow component (hundreds of ps). As summarized in Fig. 4d, the slow rise times are around 200 ps in the region (i), similar to the fitted results of D1⁺. The rise time shows a significant slowdown with the further doping increase in region (ii) and becomes comparable to the decay lifetime. The flattened rising edge for the −3 V and −3.5 V traces in Fig. 4a results from a rise and decay of similar time scales. Figure 4b, c shows the zoomed-in TR-PL of the −2 V and −3 V measurements, and the fittings with different rise times are plotted for comparison.

The lack of significant decay lifetime changes over this range indicates that no specific doping-activated relaxation channels contribute to the spectral changes in D2⁺. When considering the origin of the long rise time component, we first note that the photoexcited electrons into the optically-allowed upper conduction bands should relax to the bottom conduction bands within a few ps from bright exciton PL lifetime. These dark trions are also the energy ground states with no known lower-energy exciton reservoir. The exciton rise time of similar time scales has been associated with the momentum relaxation of excitons in previous studies in GaAs-based quantum wells[37]. Our results, therefore, suggest that there is a slowing down and blockage in D2⁺ momentum relaxation or exciton formation across the critical density $n_c$. The linewidth broadening can arise from increased phonon scattering during the slow relaxation/formation process and the blockage above $n_c$ leads to the decrease in D2⁺ amplitude.

To further verify the origin of D2⁺ spectral changes at higher doping, circularly-polarized excitations were used to compare the D2⁺ doping dependence with and without the presence of D1⁺. The valley-selective dark trion excitation pathway is explained in detail in the SI. With σ+ excitation, K valley bright excitons are excited and create a higher electron population in the K' valley through intervalley scattering, which leads to a higher D1⁺, K' valley dark trion population (Fig. 4f). With σ− excitation, it directly excites K' valley bright exciton and favors the formation of D2⁺ despite the lower hole population in the K' valley (Fig. 4g). When comparing results with σ− polarization (D2⁺ excitation) and linear polarization excitation (D1⁺ and D2⁺ excitation), the D2⁺ linewidth broadening showed a delayed onset of 0.2 V (see SI for the fitted linewidths) and a more gradual amplitude decrease. With the further doping increase, the energy separation between the D1⁺ and D2⁺ continues to decrease due to their different energy redshifts, which can lead to the broadening of D2⁺. Combined with the TR-PL discussed above, we can infer that the Fermi sea fluctuations in the K valley valence band due to D1⁺ contribute to the broadening of D2⁺. These two dark states have no direct coupling within the single-particle picture. However, the formation and recombination of D2⁺ will require a hole carrier within the same Fermi sea that dresses the D1⁺ polaron and, therefore, can be affected above

$n_c$ when correlations start to dominate. Our results suggest a strong coupling between the different trion and exciton-polaron states mediated by the Fermi sea fluctuations.

## Discussion

In summary, we reveal complex interactions between the dark trion and dark exciton-polaron states in monolayer $WSe_2$. By tracing the evolution of exciton screening as a function of charge and exciton densityies we identified a critical density $n_c$ of $-1 \times 10^{12}/cm^2$ as an onset of enhanced many-body interactions of the Fermi sea carriers for the dilute exciton limit. This crossover behavior at such a high density of $n_c$ is beyond the current exciton-polaron model and may imply additional electronic correlation effects such as Wigner crystals[38,39]. The Fermi sea screening of excitons is investigated with varying exciton density, which provides insights into the exciton-polaron modeling with exciton-exciton interaction. We also show a strong intervalley coupling between the dark trions and exciton-polarons that is mediated by the many-body interactions of the Fermi sea fluctuations. Our results lay the foundation for understanding the exciton-polaron effects after incorporating the exciton-exciton interactions, which will be important for probing excitonic phase transitions, e.g., Bose-Einstein condensation[40], in 2D systems.

## Methods

### Synthesis of WSe₂ crystals

$WSe_2$ single crystals were synthesized by a self-flux method using Se as the flux. W powder (99.999%, Alfa Aesar 12973) and Se shot (99.9999%, Alfa Aesar 10603) were sealed in a quartz ampoule under vacuum ($-10^{-6}$ Torr). Subsequently, the ampoules were heated to 1080 °C over 12 h, held there for 2 weeks, and cooled to 500 °C over 2 weeks. From 500 °C, the samples were cooled to room temperature °C over 2 days. Subsequently, the resulting boule of Se flux and $WSe_2$ single crystals were reloaded into a new quartz ampoule with alumina wool acting as a filter for the Se flux. The ampoules were heated to 300 °C and centrifuged. The then isolated single crystals were removed and sealed a third time in quartz and annealed in a temperature gradient ($T_{hot} = 275$ °C, $T_{cold} = 100$ °C) with crystals on the hot end for 2 days to remove any remaining excess Se.

### Sample and device fabrication

The measured monolayer $WSe_2$ was encapsulated with hBN, and top and bottom graphite layers were used as electrodes. An additional stripe of graphite was attached to the $WSe_2$ for grounding. All layered materials were first mechanically exfoliated from their bulk crystals onto $SiO_2$/Si substrates and identified by their color contrast under a white light optical microscope. The 2D material stack was built with the dry transfer technique using a PC stamp and released onto a substrate with pre-patterned gold electrodes.

### Measurements

Fourier plane imaging is achieved by placing an additional Fourier lens (-1 m focal length) in the collection path to project the back focal plane onto the spectrometer CCD. Details of the setup and typical results are shown in the Supplementary Information. The laser excitation was the second harmonic of an optical parametric oscillator output from Compact OPO pumped by Coherent Chameleon, with a repetition rate of 78 MHz and pulse duration of 200 fs. Time-resolved PL was measured with an avalanche photodiode (PicoQuant, PDM) coupled to a time-correlated single photon counter (HydraHarp 400). The collected PL signal was first sent into the spectrometer (Teledyne Princeton Instruments, SpectraPro HRS 300) and was filtered in both photon energy and emission momentum before the time-resolved PL measurements. The decay and rise lifetimes are fitted with the convolution of the instrument response function (-40 ps FWHM). All measurements were carried out in a closed-cycle optical cryostat (attoDRY 1000) with a base temperature of 4 K and a superconducting magnet up to 9 T.

## Data availability

The data that support the findings of this study are available within the paper and its Supplementary Information. Additional data are available from the corresponding authors upon request.

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

## Acknowledgements

X.-X.Z. acknowledges the support from National Science Foundation (NSF) award DMR-2142703. This work was partly conducted at the Research Service Centers of the Herbert Wertheim College of Engineering at the University of Florida. D.R. was supported by the University of Wisconsin-Madison, Office of the Vice Chancellor for Research and Graduate Education with funding from the Wisconsin Alumni Research Foundation.

## Author contributions

X.-X.Z. designed the study. X.C. developed the spectroscopy setup. X.C. and P.A.M. performed the measurements. X.C. and M.Z. fabricated the devices. D.R. grew the bulk WSe₂ crystals. K.W. and T.T. grew the bulk hBN crystals. X.-X.Z. wrote the manuscript. All authors discussed the results and commented on the manuscript.

## Competing interests

The authors declare no competing interests.
