## [Peer Review File · Nature Communications]

Reviewers' Comments:

Reviewer #1:

Remarks to the Author:

The paper presents a very nice experiment that demonstrates the effect of excitation power on the crossover behavior of two dark-exciton polarons in monolayer WSe₂ versus doping density under in-plane magnetic field. The analyses appear to be carefully done. Such crossover behavior may imply some many-body effects in exciton-Fermi-sea coupled system in two-dimensional materials which are still not well understood. The experiment provides many useful hints to motivate future theoretical studies on how to bridge the strongly correlated phases with Fermi-liquid phase in such a system. I would recommend the paper to be published in Nature Communications after the author consider the following questions to improve the readability of the paper.

Q1. What is basis for the assignment of D1+ and D2+ trion peaks? In the supplementary information (SI), it states that the σ^+ excitation favors the formation of D1+, which corresponds to the K' valley dark exciton that interacts with holes in the K valley valence band. σ^- excitation favors the formation of D2+, which is the K valley dark trion state. For experts familiar with the dark exciton recombination mechanism, this clue may be enough. However, for the general reader, more detailed explanation is still needed. More clarification statements should be added either in the main text or SI. It would also be nice if the assignment can be confirmed by detecting the polarization of the emitted light.

Q2. It is not clear what is the final state of system when the exciton in the D1+ or D2+ trion recombines. Fig. 1c suggests that the D1+ (D2+) trion is formed by a K' (K) dark exciton coupled to a hole in the different valley. I would assume that the recombination of the exciton in the D1+ (D2+) trion recombines and leaves behind a hole in the different valley, i.e. K (K') valley. Is this the case? It will be nice to make it clear, so the readers can understand why is the D2+ peak appears at lower energy than the D1+ peak.

Q3. I would guess that the crossover point is near the onset when the Zeeman-effect lifted valley begins to be populated by injected carriers. It is interesting to see that such a crossover point moves up to large gate voltage (in absolute magnitude) as the excitation power increases. It is hard to image that the Zeeman splitting would change significantly with the excitation power. Thus, it would be helpful if the authors can comment on other possible physical mechanisms that may cause such an effect.

Reviewer #2:

Remarks to the Author:

In this manuscript, the authors report the evolution of Fermi sea screening with increasing exciton density and the impacts of polaron-polaron interactions by measuring the p-doping dependent photoluminescence in monolayer WSe₂. The crossover of trion and exciton-polaron pictures is a topic attracted more and more attention. This manuscript used a magnetic field to separate two valleys and get the evidence to show the crossover happened at a critical carrier density. The results are interesting and I could recommend this manuscript after the authors address the following comments:

1. When mentioning the Supplementary Information in the maintext, it's better to point out which figure it is.
2. The crossover of the trion and exciton-polaron pictures in the bright excitons has been discussed in previous work (Phys. Rev. B 98, 235203, 2018, Phys. Rev. B 101, 205409, 2020, arXiv:2006.04895), the peak energies of absorption and PL separate at a critical carrier density, which is attributed to the roton effect. The authors could discuss this and compare it with their own results.
3. For the above reason, the authors could show the zero field PL and absorption results in the Supplementary Information together.
4. Is it possible to see similar phenomenon as Fig.2a with a continuous wave laser?
5. Will the intervalley dark trion (Nat. Commun. 11, 618, 2020, Phys. Rev. Lett. 124, 196802, 2020) have some kind of influence on this crossover, since the intervalley dark trion have a very

different recombination process compared with spin dark trion?

Reviewer #3:

Remarks to the Author:

The manuscript by Zhang et al. explores the doping dependence of various resonances appearing in the optical spectrum of monolayer WSe₂. This material is of high current interest, due to its large binding energies of excitons and trions, which can even be stable at room temperature. These materials furthermore have a strong coupling to light and rich valley and spin degrees of freedom, making them ideal candidates for future applications in electronics and optoelectronics.

The paper focusses on states that are connected (in the limit of vanishing doping) to dark excitons and trions. The authors demonstrate that they can isolate the signal from these optically dark states using Fourier plane imaging spectroscopy. Using this technique, they obtain exceptionally clean spectra that show an intriguing two-segment behavior of the trion energies as a function of gate voltage. The transition between the two regimes is then investigated by changing the exciton number as well as a magnetic field.

The paper is generally well written, and I found the results very intriguing. The quality of the data is exceptional. Before making a final recommendation on the paper, I feel that the authors should address the following points:

(1) If I understand well, the authors claim that the D1+ resonance changes between being due to individual trions (when the exciton density is \sim the electron density) to exciton polarons (when the exciton density $<$ electron density). While this makes intuitively sense, it is not clear to me what the authors think happens for the D2+ resonance? In particular, why does the D2+ resonance suddenly broaden, and why is this at the same gate voltage as the change in D1+? One thing that is never mentioned in the manuscript is when the K' valence band starts being occupied at $B=9T$, could this at all play a role?

(2) The authors also have a statement on page 5 that the "onset of D1+ slope change and D2+ broadening is similar only for linearly polarized excitation". I have three questions in this regard. (1) is there data for other polarizations, and if so could it be included? (2) does this data (if it exists) support the authors' hypothesis for the origin of the two-segment behavior? and (3) could the authors make it clearer elsewhere in the manuscript that they generally consider linear polarization?

(3) I am struggling to understand the significance of Fig 3 with respect to the rest of the manuscript. Could the authors explain better how Fig 3 connects to the results presented elsewhere in the manuscript? Does it support the hypothesis for the two-segment behavior or is it not connected?

(4) I believe that panel 1C is misleading: My understanding is that the Zeeman shift caused by the magnetic field shifts all "red" bands up, and "blue" bands down. However, this ordering is not respected by the lower conduction bands. Is this a mistake, or am I missing something?

(5) On page 5, the authors state "With a high doping density of $1 \times 10^{13}/\text{cm}^2$..." Should this have been exciton density rather than doping density?

(6) I do not understand the authors' statement that the "redshift in D1+ is comparable to the ... hexciton state" (page 4-5). The hexciton is a proposal for describing an exciton dressed by two electrons and two holes, which is very different from the authors' arguments that D1+ is a polaron due to the dressing of a dark exciton.

(7) I found the last sentence of the conclusion confusing. On the one hand, the authors talk about Wigner crystals, i.e., strongly correlated electron physics which does not require multiple excitons. On the other hand, they talk about exciton-exciton interactions and Bose-Einstein condensation,

which require multiple excitons. It would probably make sense to split these two parts of the outlook.

Response to Reviewers

We thank the reviewers for their insightful comments and suggestions (in blue). We addressed the questions point-by-point and revised the manuscript accordingly.

REVIEWER COMMENTS

Reviewer #1 (Remarks to the Author):

The paper presents a very nice experiment that demonstrates the effect of excitation power on the crossover behavior of two dark-exciton polarons in monolayer WSe₂ versus doping density under in-plane magnetic field. The analyses appear to be carefully done. Such crossover behavior may imply some many-body effects in exciton-Fermi-sea coupled system in two-dimensional materials which are still not well understood. The experiment provides many useful hints to motivate future theoretical studies on how to bridge the strongly correlated phases with Fermi-liquid phase in such a system. I would recommend the paper to be published in Nature Communications after the author consider the following questions to improve the readability of the paper.

Q1. What is basis for the assignment of D1+ and D2+ trion peaks? In the supplementary information (SI), it states that the σ^+ excitation favors the formation of D1+, which corresponds to the K' valley dark exciton that interacts with holes in the K valley valence band. σ^- excitation favors the formation of D2+, which is the K valley dark trion state. For experts familiar with the dark exciton recombination mechanism, this clue may be enough. However, for the general reader, more detailed explanation is still needed. More clarification statements should be added either in the main text or SI. It would also be nice if the assignment can be confirmed by detecting the polarization of the emitted light.

Response to Q1:

We fully agree with the reviewer that additional arguments/discussion on the dark trion assignment will greatly improve the clarity and readability of this manuscript. The valley index and trion configuration are assigned based on their measured g-factors and the circularly polarized excitation measurements. Specifically, the g-factor was measured to be 10 in region (i), the same as the neutral dark exciton. This indicates the relaxation pathway for these intervalley dark trions, e.g., K valley dark exciton +K' valley hole, corresponding to the recombination of the K valley exciton and leaving behind a hole in the K' valley. The other possible dark trion configuration will involve the binding of two holes in the same valence band, which will be of much higher energy due to the large exchange interactions. In this measurement, the collected dark exciton emission has an out-of-plane dipole and is z polarized, which is along the light propagation direction (e.g. see). As a result, optical components in the collection path (along the out-of-plane z direction) can only measure light polarization within the xy plane and are insensitive to the circular polarization of the dark exciton emission.

In response to this comment, we added comments in the Fig. 1c caption on the assignment of valley and circular polarization and added discussion on the assignment justification based on their Zeeman energy shift and trion configurations on Page 3 of the main text in the revised manuscript.

Q2. It is not clear what is the final state of system when the exciton in the D1+ or D2+ trion recombines. Fig. 1c suggests that the D1+ (D2+) trion is formed by a K' (K) dark exciton coupled to a hole in the different valley. I would assume that the recombination of the exciton in the D1+ (D2+) trion recombines and leaves behind a hole in the different valley, i.e. K (K') valley. Is this the case? It will be nice to make it clear, so the readers can understand why is the D2+ peak appears at lower energy than the D1+ peak.

Response to Q2:

Thank you for pointing out the potential confusions to the readers. We have added additional discussion in the main text to explain the assignment and relaxation of the different dark excitons and dark trions. The revised manuscript now includes a statement on the relaxation pathway of the dark trions on Page 3 of the main text.

Q3. I would guess that the crossover point is near the onset when the Zeeman-effect lifted valley begins to be populated by injected carriers. It is interesting to see that such a crossover point moves up to large gate voltage (in absolute magnitude) as the excitation power increases. It is hard to image that the Zeeman splitting would change significantly with the excitation power. Thus, it would be helpful if the authors can comment on other possible physical mechanisms that may cause such an effect.

Response to Q3:

We very much thank the reviewer for pointing out the potential confusions. From the performed magnetic field dependence of the crossover point V_c , as shown in Figure 2(f), we observed a lack of magnetic field dependence of the crossover point, and concluded that the crossover behavior is not directly determined by the magnetic field, as long as the Zeeman splitting is large enough so that there is no substantial spectral overlap between the two valley-polarized dark trion peaks. In comparison, if the crossover onset were determined by when both valley begin to be populated, the crossover point would show a strong field dependence, similar to the observations in Fig. 3 b & c for the bright states.

The estimated carrier densities in the K and K' bands at the crossover point also indicate the crossover does not occur along with onset of the Zeeman-split bands population. The crossover point (at low fluence) of $\sim -2V$ correspond to $1.4 \times 10^{12}/cm^2$ carrier injection. Taking the valance band (VB) g-factor of ~ 12 , with an out-of-plane 9T field, the K and K' valley valance bands have a Zeeman splitting of 6.5 meV. We can then estimate based on $0.4m_e$ effective mass of the hole carriers, that the Fermi-level touches both VBs at $\sim -0.9V$ ($6.5 \times 10^{11}/cm^2$), and the crossover occurs when the Fermi level is $\sim 4 meV$ into both bands. At the crossover, the carrier density in the K valley (following the schematics in Fig. 1c) reaches $1 \times 10^{12}/cm^2$, and the K' valley carrier density is around $4 \times 10^{11}/cm^2$.

As we increase the laser fluence to increase the K' valley dark trion density (K' valley dark exciton + K valley hole), then crossover density in K valley also increases (shown in Fig. 2e as an overall shift of crossover gate voltage). This demonstrate the dynamic screening effects of carriers with the increasing density of exciton/polaron states, and is overall consistent with the trend that more carriers are needed to fully screen out exciton states when their density is well beyond dilute limit (in comparison to the carrier density). However, further theoretical studies are still needed to fully understand the origin of the crossover onset density $1 \times 10^{12}/cm^2$. It could be related to electronic phase transition, the roton effects, or the crossover between the trion-hole complex to exciton-

polaron (as comment by Reviewer 2), but possible experiment verification of the specific mechanism is beyond the scope of this paper and will be investigated in the future.

To clarify this important point, we rewrite and expand the discussion on the magnetic field dependence of the dark exciton crossover point. The new discussions are in the main text, the last paragraph on Page 5 till Page 6. We also further clarify our statements in the differences of the magnetic field dependencies when comparing to the bright trion absorption on Page 7 of the main text.

Reviewer #2 (Remarks to the Author):

In this manuscript, the authors report the evolution of Fermi sea screening with increasing exciton density and the impacts of polaron-polaron interactions by measuring the p-doping dependent photoluminescence in monolayer WSe₂. The crossover of trion and exciton-polaron pictures is a topic attracted more and more attention. This manuscript used a magnetic field to separate two valleys and get the evidence to show the crossover happened at a critical carrier density. The results are interesting and I could recommend this manuscript after the authors address the following comments:

Q1. When mentioning the Supplementary Information in the maintext, it's better to point out which figure it is.

Response to Q1:

We are very thankful for your advice! We have made the corresponding changes to specify the supplementary figure numbers in the main text discussion.

Q2. The crossover of the trion and exciton-polaron pictures in the bright excitons has been discussed in previous work (Phys. Rev. B 98, 235203, 2018, Phys. Rev. B 101, 205409, 2020, arXiv:2006.04895), the peak energies of absorption and PL separate at a critical carrier density, which is attributed to the roton effect. The authors could discuss this and compare it with their own results.

Response to Q2:

Thanks for raising these very relevant and helpful papers that also discuss the crossover behaviors. We have added the citations and discussions in the main text and a new supplementary figure (also see the answer for comment 3).

Below we provide detailed comparisons with our results:

The crossover discussed in the PRB 98,235203 paper is the transition between the four-body trion-hole complex and the exciton-polaron states in QW systems. The authors calculated the absorption, and expected the absorption amplitude and exciton-charged exciton energy splitting to change across the crossover point. The calculation is done based on QW 2D and quasi-2D systems, and not directly about TMD materials. Since the dark excitons do not show measurable absorption strength, we also cannot compare with their calculated spectra. We can still tentatively compare with their crossover prediction for 2D : $k_F a_x = 0.7 - 0.8$ with the Fermi momentum that corresponds to our measured crossover point, which yields $k_F a_x \sim 0.3 - 0.4$. They are certainly

on the same order of magnitude. However, our measured data is not sufficient to exclusively justify this crossover mechanism. Further theoretical studies on TMD materials are still needed to draw any conclusions.

PRB 101, 205409 and arxiv:2006.04895 describe the potential roton effects in TMD excitons, where the exciton energy minimum shifts to a non-zero momentum with increasing doping density. This mechanism can also potentially explain the dark exciton crossover behavior observed in this manuscript. Once the roton minimum is fully developed at a higher doping, it's expected that the spectral linewidth goes through significant broadening, PL lifetime increase, and there will be an energy splitting between the emission and absorption feature. In our measurements, we do observe the linewidth broadening beyond the crossover point. However, we cannot directly compare the PL and absorption signal of the dark excitons, as the dark excitons can hardly be directly excited due to its very small oscillator strength, and mostly form through bright excitation relaxation. The measured PL lifetimes, while showing appreciable changes across the crossover transition, is not consistent with the theoretical predictions in PRB 101, 205409. The roton effects will increase the PL decay lifetime as a result of the direct to indirect bandgap transition. In our measurements, we observed that the decay time remained *unchanged*, while the formation/rise time *increased* during the crossover. With the above reasons, especially the lack of direct absorption information, we also cannot draw a conclusion on the possible roton effects on dark excitons.

We have also added a new supplementary figure (according to comment 3) and compared the PL and absorption peak energies for the bright neutral exciton (X0) and bright p-type trions (X+), which we can indeed get the absorption signal. As shown in Fig. R1, the onset of the bright exciton absorption and PL energy splitting is at $\sim -3V$, which is consistent with the results obtained in arxiv:2006. Taking 0V as the near neutral doping position and the device conditions specified in the main text, this doping density (onset of splitting) correspond to the $\sim 1 \times 10^{12}/cm^2$, the same as the crossover transition density extracted from the dark exciton PL. This may be an interesting indication of the roton effects in the dark excitons, and we have added relevant discussion in the main text and supplementary information.

In response to this comment, we have expanded our discussion on the crossover phenomena to include the above-mentioned papers and roton effects on Page 6 of the main text, which greatly strengthened our discussions. We have also added a *new* supplementary figure (Fig. S7) and corresponding descriptions in Supplementary Section 4.

Q3. For the above reason, the authors could show the zero field PL and absorption results in the Supplementary Information together.

Response to Q3:

As explained our response in the previous question, the dark exciton absorptions cannot be measured with the current spectroscopy techniques. We plotted the comparison of the bright exciton PL and absorption in Fig. R1.

This figure is added as a new figure in the Supplementary Information.

Figure. R1. Comparison between peaks energies of the bright neutral exciton (X0) and bright p-type trion (X+) as measured from PL spectroscopy and reflection contrast spectroscopy (absorption). The plotted measurements were done at 4K, 0T.

Q4. Is it possible to see similar phenomenon as Fig.2a with a continuous wave laser?

Response to Q4:

Yes, thanks for bringing up this point. We have done thorough measurements with cw laser as well. The cw measurements showed the same trends as the lowest fluence pulsed laser excitation in the manuscript. With the cw laser source, we remained at a low dark exciton density and did not observe the fluence dependent shifting of the crossover point, consistent with our expectation.

We added the statements on the cw laser results on Page 5 of the main text.

Q5. Will the intervalley dark trion (Nat. Commun. 11, 618, 2020, Phys. Rev. Lett. 124, 196802, 2020) have some kind of influence on this crossover, since the intervalley dark trion have a very different recombination process compared with spin dark trion?

Response to Q5:

The p-type dark trions we discussed in the manuscript have the intervalley configuration, e.g. K valley spin-dark exciton + K' valence band hole. From the papers cited in this comment, we assumed that the referee intended to ask about the phonon assisted relaxations of dark excitons, and therefore the phonon replicas. The Γ_5 phonon replica can be observed in our data using a low laser fluence, which was discussed in Supplementary Information Section 4, and Fig. S6. The phonon replicas of each of the Zeeman-split dark trions show the same energy redshift and gate dependence. However we cannot do the same fluence dependence measurements with these phonon replicas because they were difficult to extract with the superlinear growth of biexciton emissions (at similar energies) at higher excitation densities.

Reviewer #3 (Remarks to the Author):

The manuscript by Zhang et al. explores the doping dependence of various resonances appearing in the optical spectrum of monolayer WSe₂. This material is of high current interest, due to its large binding energies of excitons and trions, which can even be stable at room temperature. These materials furthermore have a strong coupling to light and rich valley and spin degrees of

freedom, making them ideal candidates for future applications in electronics and optoelectronics.

The paper focusses on states that are connected (in the limit of vanishing doping) to dark excitons and trions. The authors demonstrate that they can isolate the signal from these optically dark states using Fourier plane imaging spectroscopy. Using this technique, they obtain exceptionally clean spectra that show an intriguing two-segment behavior of the trion energies as a function of gate voltage. The transition between the two regimes is then investigated by changing the exciton number as well as a magnetic field.

The paper is generally well written, and I found the results very intriguing. The quality of the data is exceptional. Before making a final recommendation on the paper, I feel that the authors should address the following points:

Q1. If I understand well, the authors claim that the D1+ resonance changes between being due to individual trions (when the exciton density is \sim the electron density) to exciton polarons (when the exciton density $<$ electron density). While this makes intuitively sense, it is not clear to me what the authors think happens for the D2+ resonance? In particular, why does the D2+ resonance suddenly broaden, and why is this at the same gate voltage as the change in D1+? One thing that is never mentioned in the manuscript is when the K' valence band starts being occupied at B=9T, could this at all play a role?

Response to Q1:

Thank you very much for pointing out the potential confusions in our manuscript. The D2+ gate dependence was discussed on page 7&8 of the main text. To summarize and more explicitly answer the referee's question, the D2+ shows broadening as the D1+ exciton transits into exciton-polaron state. Beyond the crossover point, D1+ (K' valley dark exciton + K valley hole) experience stronger screening and is dressed with electron-hole pair excitations in the K valley holes, following the schematics in Fig. 4e. The onset of this strong polaron effect gives rise to a significant slowdown in the rise time of D2+ (Fig. 4), which indicates an increase in dark exciton formation or relaxation process. The longer formation and relaxation time involve more scattering of this D2+ exciton and therefore gives rise to broader spectral width.

We qualitatively understood it as the presence of electron-hole excitations in the K valley Fermi sea hinders the emission of the D2+, which involves K valley spin dark exciton. This is also confirmed with the comparison with circularly polarized excitation: when exciting into D2+ state (with lower D1+ population), the onset of D2+ linewidth broadening was delayed, as shown in Supplementary Figure S5. However, we cannot exclusively identify the microscopic origin, which will require a future theoretical studies.

For the question about K' valence band occupation: at 9T, the K' valence band starts to be populated at -0.9V. At the crossover of -2V (for low fluences), the K valence band carrier density is around $1 \times 10^{12}/\text{cm}^2$, and the K' valley carrier density is around $4 \times 10^{11}/\text{cm}^2$. Please also refer to our answer to Reviewer 1, Q3, where we list the detailed parameters used in this estimation. Given the lack of magnetic field dependence of the crossover voltage, as shown in the main text Figure 2f, we don't think the crossing of Fermi levels of the K and K' valley plays a big role, which is contrary to the case of bright exciton absorption (Figure 3).

We have added discussions on this point in the manuscript on Pages 5-6 of the main text to clarify this important point. We also further clarify our statements in the differences of the magnetic field dependencies when comparing to the bright trion absorption on Page 7 of the main text.

Q2 The authors also have a statement on page 5 that the "onset of D1+ slope change and D2+ broadening is similar only for linearly polarized excitation". I have three questions in this regard. (1) is there data for other polarizations, and if so could it be included? (2) does this data (if it exists) support the authors' hypothesis for the origin of the two-segment behavior? and (3) could the authors make it clearer elsewhere in the manuscript that they generally consider linear polarization?

Response to Q2:

(1) The data with circularly polarized excitation is shown in the main text Figure 4f, discussed on page 8 of the main text (the paragraph before the discussion), with further analysis of the linewidth broadening shown in the Supplementary Information Figure S4 and S5, and in Section 3 "Valley polarization of dark trions and intervalley coupling". One thing to note is that because the collected dark exciton emission comes from the out-of-plane dipole and is Z-polarized, the signal does not carry circular polarization from the valley-spin coupling. As a result, we only performed circularly-polarized excitation measurements.

(2) With circularly polarized excitation, we can detect the doping response of the two different dark trions (D1+ and D2+) while modulating their population ratio. As discussed on page 8 and Supplementary Section 3, we can observe a delay in D2+ crossover (through linewidth broadening) with fewer D1+ population even under the same doping. These circularly polarized experiments demonstrate the many-body nature of the crossover and indicates the presence of intervalley coupling that is solely mediated by the Fermi sea fluctuations. The results provide us with crucial information to understand our results.

(3) Yes. Thank you very much for raising this important point. We have added on Page 2 of the main text to clarify the excitation conditions of the experiments.

Q3. I am struggling to understand the significance of Fig 3 with respect to the rest of the manuscript. Could the authors explain better how Fig 3 connects to the results presented elsewhere in the manuscript? Does it support the hypothesis for the two-segment behavior or is it not connected?

Response to Q3:

Thank you very much for pointing out this potential confusion. Figure 3 of the main text, which shows the magnetic field dependence of the bright exciton and trion absorption, provides crucial information that exclude the possible causes/mechanism of the two-segment dark trion gate dependence. The dark trions, which is the main topic of this manuscript, are formed through bright exciton light absorption and subsequent intervalley carrier scattering. From the absorption spectra in Fig. 3, it can be seen that two-segment in the dark trion emission does not simply follow the gate dependence of the bright exciton absorption, and therefore not a result of the absorption changes, including Pauli blocking effects, effective mass changes and the Landau level formations. In particular, the bright trion absorption also shows a different two-segment gate dependence, which might come from the effective mass change when the Fermi level touches the both of the Zeeman-split valence bands (see PhysRevX.10.021024 (2020), Nature Nano 12, 144, doi:10.1038/nano.2016.213). As also discussed in reply to comment 1, the comparison of Fig. 3c and Fig. 2f is to demonstrate the lack of magnetic field dependence in the dark trion crossover, and exclude the potential roles of K valley occupation the referee talked about in comment 1.

We partially rewrite the description for Figure 3, which is on Page 7 of the main text. In the revised manuscript, we emphasize how the comparison to bright trion absorptions assist the analysis of bright polaron crossover, in terms of excluding possible mechanism and verifying the magnetic field dependence.

Q4. I believe that panel 1C is misleading: My understanding is that the Zeeman shift caused by the magnetic field shifts all "red" bands up, and "blue" bands down. However, this ordering is not respected by the lower conduction bands. Is this a mistake, or am I missing something?

Response to Q4:

Thank you for pointing out potential confusions. The plotted different conduction and valence band Zeeman energy splittings in Fig. 1c are sketched based the different g-factors of the bands. The spin, orbital angular momentum and valley Berry curvature gives different contributions to their g-factors. The orbital contribution is much higher for the valence band. The lowest conduction band here, despite having the opposite spin g-factor, has slightly higher contributions from the *opposite valley Berry curvature magnetic momentum*. The valley contribution of magnetic momentum gives the same sign for the Zeeman shifts of the lowest conduction and highest valence band for monolayer WSe₂.

In monolayer WSe₂, the valence band g-factor is reported to be ± 6 , and the lowest conduction band (relevant for the spin dark exciton) g-factor of ± 0.9 from "Measurement of conduction and valence bands g-factors in a transition metal dichalcogenide monolayer." PRL126.6 (2021): 067403. Figure 2 of this PRL paper also depicts the band shifting of WSe₂, consistent with our figures. The different contributions of the conduction and valence band g-factors are also discussed in a number of theoretical papers like "Orbital, spin and valley contributions to Zeeman splitting of excitonic resonances in MoSe₂, WSe₂ and WS₂ Monolayers." 2D Materials 6.1 (2018): 015001.

We have added the citation of "Measurement of conduction and valence bands g-factors in a transition metal dichalcogenide monolayer" when we first introduce the schematics of electronic Zeeman splitting on Page 3 of the main text.

Q5. On page 5, the authors state "With a high doping density of $1 \times 10^{13}/\text{cm}^2$..." Should this have been exciton density rather than doping density?

Response to Q5:

Thank you very much for the correction! It is indeed the exciton density and not the doping density. We have corrected this in the revised manuscript.

Q6. I do not understand the authors' statement that the "redshift in D1+ is comparable to the ... hexicton state" (page 4-5). The hexicton is a proposal for describing an exciton dressed by two electrons and two holes, which is very different from the authors' arguments that D1+ is a polaron due to the dressing of a dark exciton.

Response to Q6:

We fully agree that these D1+ polaron the theoretically proposed hexicton states are very different quasiparticles and have now removed this statement. We compared with that specific PL signal (assigned to hexicton) to have a qualitative understanding the energy redshifts as a function of doping observed in the D1+ polaron. In prior studies of the bright exciton and bright exciton

polaron, the energy separation between the bright neutral and trion states (attractive and repulsive polaron) as a function of doping (therefore Fermi energy) has been used as a hallmark of the 2D Fermi-polaron. In the case of dark excitons and dark trions, we cannot extract such a quantity due to (1) neutral dark exciton PL signal quenches immediately away from the neutral point (2) dark excitons are not observable in optical absorption. The proposed hexciton state in PRL 129, 076801 (2022) has energy redshift with doping densities due to the phase space filling effects on the quasiparticle's binding energy, and the similar energy redshift slope in the D1+ could suggest a similar many-body effect.

We agree that the comparison is not rigorous and necessary here, and we have thus removed this statement in the revised manuscript.

Q7. I found the last sentence of the conclusion confusing. On the one hand, the authors talk about Wigner crystals, i.e., strongly correlated electron physics which does not require multiple excitons. On the other hand, they talk about exciton-exciton interactions and Bose-Einstein condensation, which require multiple excitons. It would probably make sense to split these two parts of the outlook.

Thank you very much for pointing out this important distinction. We agree that these are two different physical interactions and need to be discussed separately.

We have split this into a two-part statement in the resubmitted version.

Reviewers' Comments:

Reviewer #1:

Remarks to the Author:

The authors have improved the manuscript according to my suggestions, and I am satisfied with the revisions. The manuscript is much more readable now, and I recommend its publication for Nature Communications.

Reviewer #2:

Remarks to the Author:

The revised manuscript has addressed all my concerns. I would thus recommend the publication of the manuscript in Nature Communications.

Reviewer #3:

Remarks to the Author:

Having read the other referees' reports as well as the authors' very clear responses to all queries and associated careful changes of the manuscript, I am happy to recommend publication in Nature Communications.

REVIEWERS' COMMENTS

Reviewer #1 (Remarks to the Author):

The authors have improved the manuscript according my suggestions, and I am satisfied with the revisions. The manuscript is much more readable now, and I recommend its publication for Nature Communications.

Response: We are very grateful for the helpful suggestions from reviewers, which improved the readability and clarity of our manuscript!

Reviewer #2 (Remarks to the Author):

The revised manuscript has addressed all my concerns. I would thus recommend the publication of the manuscript in Nature Communications.

Response: We are very glad that our previous response addressed your concerns. Thank you for the helpful comments!

Reviewer #3 (Remarks to the Author):

Having read the other referees' reports as well as the authors' very clear responses to all queries and associated careful changes of the manuscript, I am happy to recommend publication in Nature Communications.

Response: Thank you very much for your recommendation. We really appreciate your review comments and suggestions, which helped improve the clarity of the manuscript.